# Learning to cluster in order to transfer across domains and tasks

**Yen-Chang Hsu, Zhaoyang Lv**
Georgia Institute of Technology
Atlanta, GA 30332, USA
`{yenchang.hsu, zhaoyang.lv}@gatech.edu`

**Zsolt Kira**
Georgia Tech Research Institute
Atlanta, GA 30318, USA
`zkira@gatech.edu`

## Abstract

This paper introduces a novel method to perform transfer learning across domains and tasks, formulating it as a problem of learning to cluster. The key insight is that, in addition to features, we can transfer similarity information and this is sufficient to learn a similarity function and clustering network to perform both domain adaptation and cross-task transfer learning. We begin by reducing categorical information to pairwise constraints, which only considers whether two instances belong to the same class or not (pairwise semantic similarity). This similarity is category-agnostic and can be learned from data in the source domain using a similarity network. We then present two novel approaches for performing transfer learning using this similarity function. First, for unsupervised domain adaptation, we design a new loss function to regularize classification with a constrained clustering loss, hence learning a clustering network with the transferred similarity metric generating the training inputs. Second, for cross-task learning (i.e., unsupervised clustering with unseen categories), we propose a framework to reconstruct and estimate the number of semantic clusters, again using the clustering network. Since the similarity network is noisy, the key is to use a robust clustering algorithm, and we show that our formulation is more robust than the alternative constrained and unconstrained clustering approaches. Using this method, we first show state of the art results for the challenging cross-task problem, applied on Omniglot and ImageNet. Our results show that we can reconstruct semantic clusters with high accuracy. We then evaluate the performance of cross-domain transfer using images from the Office-31 and SVHN-MNIST tasks and present top accuracy on both datasets. Our approach doesn't explicitly deal with domain discrepancy. If we combine with a domain adaptation loss, it shows further improvement.

## 1 Introduction

Supervised learning has made significant strides in the past decade, with substantial advancements arising from the use of deep neural networks. However, a large part of this success has come from the existence of extensive labeled datasets. In many situations, it is not practical to obtain such data due to the amount of effort required or when the task or data distributions change dynamically. To deal with these situations, the fields of transfer learning and domain adaptation have explored how to transfer learned knowledge across tasks or domains. Many approaches have focused on cases where the distributions of the features and labels have changed, but the task is the same (e.g., classification across datasets with the same categories). Cross-task transfer learning strategies, on the other hand, have been widely adopted especially in the computer vision community where features learned by a deep neural network on a large classification task have been applied to a wide variety of other tasks (Donahue et al., 2014).

Most of the prior cross-task transfer learning works, however, require labeled target data to learn classifiers for the new task. If labels of the target data are absent, there is little choice other than to apply unsupervised approaches such as clustering on the target data with pre-trained feature representations. In this paper, we focus on the question of what can be transferred (besides features) to support both cross-domain and cross-task transfer learning. We address it with a learned similarity function as the fundamental component of clustering. Clustering can then be realized using a neural

network trained using the output of the similarity function, which can be successfully used to achieve both cross-task and cross-domain transfer.

The key idea is to formulate the clustering objective to use a learnable (and transferable) term, which in our proposed work is a similarity prediction function. Our proposed objective function can be easily combined with deep neural networks and optimized end-to-end. The features and clustering are optimized jointly, hence taking advantage of such side information in a robust way. Using this method, we show that unsupervised learning can benefit from learning performed on a distinct task, and demonstrate the flexibility of further combining it with a classification loss and domain discrepancy loss.

In summary, we make several contributions. First, we propose to use predictive pairwise similarity as the knowledge that is transferred and formulate a learnable objective function to utilize the pairwise information in a fashion similar to constrained clustering. We then provide the methodologies to deploy the objective function in both cross-task and cross-domain scenarios with deep neural networks. The experimental results for cross-task learning on Omniglot and ImageNet show that we can achieve state of the art clustering results with predicted similarities. On the standard domain adaptation benchmark Office-31 dataset, we demonstrate improvements over state-of-art even when not performing any explicit domain adaptation, and further improvements if we do. Finally, on another domain adaptation task, SVHN-to-MNIST, our approach using Omniglot as the auxiliary dataset achieves top performance with a large margin.

## 2 RELATED WORK

**Transfer Learning:** Transfer learning aims to leverage knowledge from the source domain to help learn in the target domain, while only focusing on the performance on the target domain. The type of transferred knowledge includes training instances, features, model parameters and relational knowledge (Pan & Yang, 2010). Pairwise similarity is the meta-knowledge we propose to transfer, which falls in between the last two types. The similarity prediction function is a neural network with learned parameters, while the output is the most simplified form of relational knowledge, i.e., only considering pairwise semantic similarity.

**Cross-task Transfer Learning:** Features learned when trained for ImageNet classification (Russakovsky et al., 2015) have boosted the performance of a variety of vision tasks in a supervised setting. For example, new classification tasks (Donahue et al., 2014; Yosinski et al., 2014), object detection (Girshick et al., 2014), semantic segmentation (Long et al., 2015a), and image captioning (Vinyals et al., 2015). Translated Learning (Dai et al., 2009) has an unsupervised setting similar to ours, but it again focuses only on transferring features across tasks. Our work explores how learning could benefit from transferring pairwise similarity in an unsupervised setting.

**Cross-domain Transfer Learning:** Also known as domain adaptation (Pan & Yang, 2010), there has recently been a large body of work dealing with domain shift between image datasets by minimizing domain discrepancy (Tzeng et al., 2014; Long et al., 2015b; Ganin et al., 2016; Sun et al., 2016; Long et al., 2016; Sener et al., 2016; Carlucci et al., 2017; Long et al., 2017; Zellinger et al., 2017; Bousmalis et al., 2017). We address the problem in a complementary way that transfers extra information from the auxiliary dataset and show a larger performance boost with further gains using an additional domain discrepancy loss.

**Constrained Clustering:** Constrained clustering algorithms can be categorized by how they utilize constraints (e.g. pairwise information). The first set of work use the constraints to learn a distance metric. For example, DML (Xing et al., 2003), ITML (Davis et al., 2007), SKMS (Anand et al., 2014), SKKm (Anand et al., 2014; Amid et al., 2016), and SKLR (Amid et al., 2016). This group of approaches closely relates to metric learning and needs a clustering algorithm such as K-means in a separate stage to obtain cluster assignments. The second group of work use constraints to formulate the clustering loss. For example, CSP (Wang et al., 2014) and COSC (Rangapuram & Hein, 2012). The third group uses constraints for both metric learning and the clustering objective, such as MPCKMeans (Bilenko et al., 2004) and CECM (Antoine et al., 2012). The fourth group does not use constraints at all. A generic clustering algorithms such as K-means (MacQueen et al., 1967), LSC (Chen & Cai, 2011), and LPNMF (Cai et al., 2009) all belong to this category. There is a long list of associated works and they are summarized in survey papers, e.g. Davidson & Basu

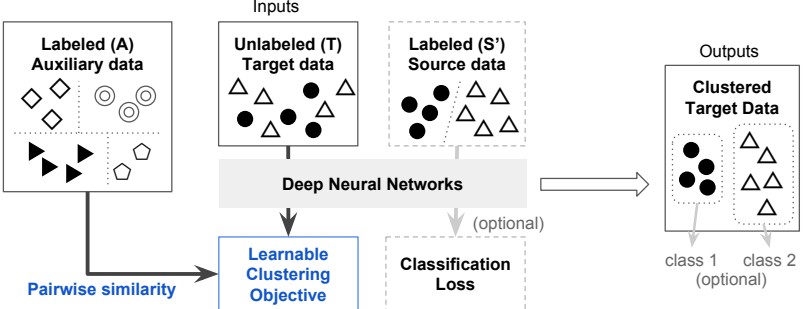

Figure 1: Overview of the transfer scheme with a learnable clustering objective (LCO). The LCO and pairwise similarity are the two key components of our approach and are described in section 4. The dashed rectangles and light gray arrows are only available in cross-domain transfer. Details are described in section 3.

(2007) and Dinler & Tural (2016). Our proposed clustering strategy belongs to the third group. The constrained clustering methods above are applied to the semi-supervised setting where the ground-truth constraints are sparsely available. In our unsupervised setting, the ground-truth is unavailable but predicted constraints are densely available. We include all four groups of algorithms in our comparison and show the advantages of the third group.

## 3 THE TRANSFER LEARNING TASKS

To define the transfer learning problem addressed in this work, we follow the notations used by Pan & Yang (2010). The goal is to transfer knowledge from source data $S = (X_S, Y_S)$, where $X_S$ is the set of data instances and $Y_S$ is the corresponding categorical labels, to a target data noted as $T = (X_T, Y_T)$. The learning is unsupervised since $Y_T$ is unknown. The scenario is divided into two cases. One is $\{Y_T\} \neq \{Y_S\}$, which means the set of categories are not the same and hence the transfer is across tasks. The second case is $\{Y_T\} = \{Y_S\}$, but with a domain shift. In other words, the marginal probability distributions of the input data are different, i.e., $P(X_T) \neq P(X_S)$. The latter is a cross-domain learning problem also called transductive learning. The domain adaptation approaches which have gained significant attention recently belong to the second scenario.

To align with common benchmarks for evaluating transfer learning performance, we further use the notion of an auxiliary dataset and split the source data into $S = S' \cup A$. $S' = (X'_S, Y'_S)$ which is only present in the cross-domain transfer scheme and has $\{Y'_S\} = \{Y_T\}$. $A = (X_A, Y_A)$ is the auxiliary dataset which has a large amount of labeled data and potentially categories as well, and may or may not contain the categories of $Y_T$. For the cross task scenario, only $A$ and unlabeled $T$ are included, while cross-domain transfer involves $A$, $S'$, and $T$. In the following sections we use the above notations and describe the two transfer learning tasks in detail. Figure 1 illustrates how our approach relates to both tasks.

**Transfer across tasks:** If the target task has different categories, we cannot directly transfer the classifier from source to target, and there is no labeled target data to use for fine-tuning transferred features. Here we propose to first reduce the categorization problem to a surrogate same-task problem. We can directly apply transductive transfer learning (Pan & Yang, 2010) to the transformed task. The cluster structure in the target data is then reconstructed using the predictions in the transformed task. See figure 2 for an illustration.

The source involves a labeled auxiliary dataset $A$ and an unlabeled target dataset $T$. $Y_T$ is the target that must be inferred. In this scenario the set of categories $\{Y_A\} \neq \{Y_T\}$, and $Y_T$ is unknown. We first transform the problem of categorization into a pairwise similarity prediction problem. In other words, we specify a transformation function $R$ such that $R(A) = (X_A^R, Y^R)$, and $X_A^R = \{(x_{A,i}, x_{A,j})\}_{\forall i,j}$ contains all pairs of data, where $\{Y^R\} = \{dissimilar, similar\}$. The transformation on the labeled auxiliary data is straightforward. It will be $similar$ if two data instances are from the same category, and vice versa. We then use it to learn a pairwise similarity prediction function $G(x_i, x_j) = y_{i,j}$.

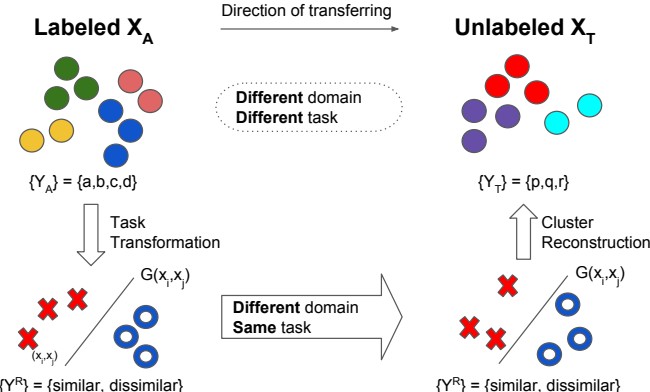

Figure 2: The concept for reconstructing clusters of unseen categories. The proposed approach follows the arrows in the counter-clockwise direction, which converts cross-task transfer learning to cross-domain transfer learning. The colors of dots represent data of different categories. The hollow circle and cross symbol represent similar and dissimilar data pairs. The $G$ function and the cluster reconstruction (via constrained clustering) are the two key components in the diagram.

By applying $G$ on $T$, we can obtain $G(x_{T,i}, x_{T,j}) = y_{T,i,j}$. The last step is to infer $Y_T$ from $Y_T^R = \{y_{T,i,j}\}_{\forall i,j}$, which can be solved using constrained clustering algorithms. Note that since the actual $Y_T$ is unknown, the algorithm only infers the indices of categories, which could be in arbitrary order. The resulting clusters are expected to contain coherent semantic categories.

**Transfer across domains:** The problem setting we consider here is the same as unsupervised domain adaptation. Following the standard evaluation procedure (Long et al., 2017; Ganin et al., 2016), the labeled datasets $A$ is ImageNet and $S'$ is one domain in the Office-31 dataset. The unlabeled $T$ is another domain in Office-31. The goal is to enhance classification performance on $T$ by utilizing $A$, $S'$, and $T$ together.

## 4 THE LEARNABLE CLUSTERING OBJECTIVE (LCO)

The key to our approach is the design of a learning objective that can use (noisy) predicted pairwise similarities, and is inspired from constrained clustering which involves using pairwise information in the loss function. The pairwise information is called must-link/cannot-link constraints or similar/dissimilar pairs (we use the latter). Note that the information is binarized to one and zero for similar and dissimilar pairs, accordingly.

Although many constrained clustering algorithms have been developed, few of them are scalable with respect to the number of pairwise relationships. Further, none of them can be easily integrated into deep neural networks. Inspired by the work of Hsu & Kira (2016), we construct a contrastive loss for clustering on the probability outputs of a softmax classifier. However, each output node does not have to map to a fixed category but instead each output node represents a probabilistic assignment of a data point to a cluster. The assignment between output nodes and clusters are formed stochastically during the optimization and is guided by the pairwise similarity. If there is a similar pair, their output distribution should be similar, and vice-versa.

Specifically, we use the pair-wise KL-divergence to evaluate the distance between $k$ cluster assignment distributions of two data instances, and use predicted similarity to construct the contrastive loss. Given a pair of data $x_p, x_q$, their corresponding output distributions are defined as $\mathcal{P} = f(x_p)$ and $\mathcal{Q} = f(x_q)$, while $f$ is the neural network. The cost of a similar pair is described as :

$$\mathcal{L}(x_p, x_q)^+ = \mathcal{D}_{KL}(\mathcal{P}^\star || \mathcal{Q}) + \mathcal{D}_{KL}(\mathcal{Q}^\star || \mathcal{P}) \tag{1}$$

$$\mathcal{D}_{KL}(\mathcal{P}^\star || \mathcal{Q}) = \sum_{c=1}^{k} p_c log(\frac{p_c}{q_c}) \tag{2}$$

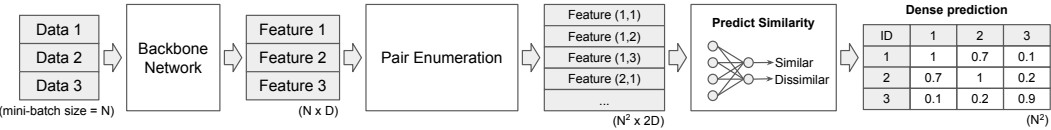

Figure 3: The similarity prediction network (the $G$ function).

The cost $\mathcal{L}(x_p, x_q)^+$ is symmetric w.r.t. $x_p, x_q$, in which $\mathcal{P}^\star$ and $\mathcal{Q}^\star$ are alternatively assumed to be constant. Each KL-divergence factor $\mathcal{D}_{KL}(\mathcal{P}^\star||\mathcal{Q})$ becomes a unary function whose gradient is simply $\partial \mathcal{D}_{KL}(\mathcal{P}^\star||\mathcal{Q})/\partial \mathcal{Q}$.

If $x_p, x_q$ comes from a pair which is dissimilar, their output distributions are expected to be different, which can be defined as a hinge-loss function:

$$\mathcal{L}(x_p, x_q)^- = L_h(\mathcal{D}_{KL}(\mathcal{P}^\star||\mathcal{Q}), \sigma) + L_h(\mathcal{D}_{KL}(\mathcal{Q}^\star||\mathcal{P}), \sigma) \tag{3}$$

$$L_h(e, \sigma) = max(0, \sigma - e) \tag{4}$$

Given a pair of data with similarity prediction function $G(x_p, x_q) \in \{0, 1\}$, which is introduced in section 3, the total loss can be defined as a contrastive loss (where we use integer 1 to represent a similar pair):

$$\mathcal{L}(x_p, x_q) = G(x_p, x_q)\mathcal{L}(x_p, x_q)^+ + (1 - G(x_p, x_q))\mathcal{L}(x_p, x_q)^- \tag{5}$$

We refer to equation 5 as LCO. Function $G$ is the learnable part that utilizes prior knowledge and is trained with auxiliary dataset $A$ before optimizing LCO. Two particular characteristics of the clustering criterion are worth mentioning: (1) There is no need to define cluster centers. (2) There is no predefined metric applied on the feature representation. Instead, the divergence is calculated directly on the cluster assignment; therefore, both feature representation and clustering are jointly optimized using back-propagation through the deep neural networks.

### 4.1 THE PAIRWISE SIMILARITY PREDICTION NETWORK

Although there is no restriction of how $G$ should be constructed, we choose deep convolution neural networks due to their efficiency in vision tasks. We design a network architecture inspired from Zagoruyko & Komodakis (2015). While they use it to predict image patch similarity, we use it to predict image-level semantic similarity. However, the Siamese architecture used in Zagoruyko & Komodakis (2015) is not efficient in both training and inference, especially when pairwise information is dense. Therefore, instead of using Siamese architecture, we keep the single column backbone but add a *pair-enumeration layer* on top of the feature extraction network. The pair-enumeration layer enumerates all pairs of feature vectors within a mini-batch and concatenates the features. Suppose the input of the layer is $10 \times 512$ with the mini-batch size 10 and the feature dimension 512; then the output of the pair-enumeration layer will be $100 \times 1024$ (self-pairs included).

The architecture is illustrated in figure 3. We add one hidden fully connected layer on top of the enumeration layer and a binary classifier at the end. We use the standard cross-entropy loss and train it end-to-end. The supervision for training was obtained by converting the ground-truth category labels into binary similarity, i.e., if two samples are from the same class then their label will be *similar*, otherwise *dissimilar*. The inference is also end-to-end, and it outputs predictions among all similarity pairs in a mini-batch. The output probability $g \in [0, 1]$ with 1 means more similar. We binarize $g$ at 0.5 to obtain discrete similarity predictions. In the following sections, we simplified the notation of the pairwise similarity prediction network as $G$. Once $G$ is learned, it then works as a static function in our experiments.

### 4.2 THE OBJECTIVE FUNCTION WITH DENSE SIMILARITY PREDICTION

Since the pairwise prediction of a mini-batch can be densely obtained from $G$, to efficiently utilize the pair-wise information without forwarding each data multiple times we also combine the pair-enumeration layer described in section 4.1 with equation 5. In this case, the outputs of softmax are

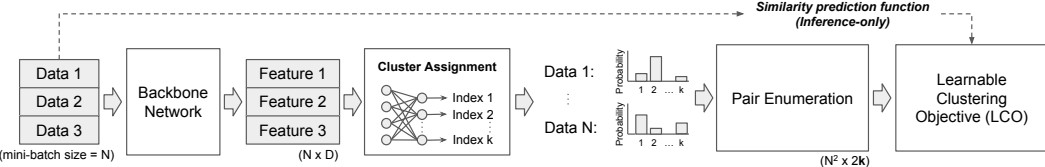

Figure 4: The constrained clustering network (CCN) for transfer learning across tasks. The input is unlabeled target data $T$. The cluster assignment block contains two fully connected layers and has the number of output nodes equal to $k$. The $f$ described in section 4 is the backbone network plus the cluster assignment block. To optimize LCO, the full pipeline in the diagram is used. After the optimization, it uses another forward propagation with only $f$ to obtain the final cluster assignment.

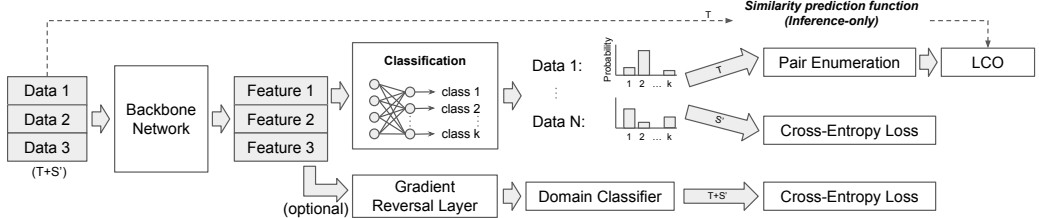

Figure 5: The network for transfer learning across domains. The input is the mix of $S'$ and $T$. The architecture is a direct extension of CCN. We use CCN⁺ to represent the mandatory parts (upper branch) which implements eq. (7). CCN⁺⁺ includes the domain adaptation method (optional branch).

enumerated in pairs. Let $D$ be the set of all tuples $(p, q)$, while $p$ and $q$ are the indices of a sample in a mini-batch. The dense clustering loss $\mathcal{L}_d$ for a mini-batch is calculated by:

$$\mathcal{L}_d = \sum_{\forall (p,q) \in D} \mathcal{L}(x_p, x_q). \tag{6}$$

We use $\mathcal{L}_d$ standalone with deep neural networks to reconstruct semantic clusters and for transfer learning across tasks (figure 4). We call the architecture the constrained clustering network (CCN).

### 4.3 COMBINING WITH OTHER OBJECTIVES

In the cross-domain case, we additionally have labeled source data. This enables us to use LCO for $T$ while using classification loss for $S'$. The overall training procedure is similar to previous domain adaptation approaches in that both source and target data are mixed in a mini-batch, but different losses are applied. We denote the source domain images in a mini-batch $b$ as $X_{b^S}$ and the target domain images $X_{b^T}$ with its set of dense pairs $D_{b^T}$. The loss function $\mathcal{L}_{cd}$ for cross-domain transfer in a mini-batch can then be formulated as:

$$\mathcal{L}_{cd} = \mathcal{L}_{cls} + \mathcal{L}_{cluster} \tag{7}$$

$$\mathcal{L}_{cls} = \frac{1}{|b^S|} \sum_{\forall x_i \in X_{b^S}} CrossEntropyLoss(f(x_i)). \tag{8}$$

$$\mathcal{L}_{cluster} = \frac{1}{|b^T|^2} \sum_{\forall (i,j) \in D_{b^T}} \mathcal{L}(x_i, x_j). \tag{9}$$

The $\mathcal{L}_{cluster}$ and $\mathcal{L}_{cls}$ share the same outputs from network $f$. Although $\mathcal{L}_{cluster}$ does not force the mapping between clusters and categories, $\mathcal{L}_{cls}$ does. Therefore the learned classifier can be applied on target data directly and picks the maximum probability for the predicted class. Note that in our loss function, there is no term to explicitly match the feature distribution between source and target; it merely transfers more knowledge in the form of constraints to regularize the learning of the classifier. There is also no requirement for the architecture to utilize hidden layers. Therefore our approach has the large flexibility to be combined with other domain adaptation strategies. Figure 5 illustrates the architectures CCN⁺ and CCN⁺⁺ used in our cross-domain experiments.

## 5 Experiments

This section contains evaluations with four image datasets and covers both cross-task and cross-domain schemes. The details are described below, and the differences between experimental settings are illustrated in appendix A.

### 5.1 Unsupervised Cross-Task Transfer Learning

#### 5.1.1 Setup

The **Omniglot** dataset (Lake et al., 2015) contains 1623 different handwritten characters and each of them has 20 images drawn by different people. The characters are from 50 different alphabets and were separated to 30 background sets $Omniglot_{bg}$ and 20 evaluation sets $Omniglot_{eval}$ by the author. We use the $Omniglot_{bg}$ as the auxiliary dataset ($A$) and the $Omniglot_{eval}$ as the target data ($T$). The total number of characters in $Omniglot_{bg}$ is 964, which can be regarded as the number of categories available to learn the semantic similarity. The goal is to cluster $Omniglot_{eval}$ to reconstruct its semantic categories, without ever having any labels.

**The $G$ function** has a backbone neural network with four 3x3 convolution layers followed by 2x2 max-pooling with stride 2. All hidden layers are followed by batch normalization (Ioffe & Szegedy, 2015) and rectified linear unit. To prepare the inputs for training, the images from $Omniglot_{bg}$ were resized to 32x32 and normalized to zero mean with unit standard deviation. Each mini-batch has a size of 100 and is sampled from a random 20 characters to make sure the amount of similar pairs is reasonable. After pair enumeration, there are 10000 pairs subject to the loss function, which is a two-class cross entropy loss. The ground-truth similarity is obtained by converting the categorical labels. The loss of $G$ is optimized by stochastic gradient descent and is the only part trained in a supervised manner in this section.

**The Constrained Clustering Network (CCN)** is used to reconstruct the semantic clusters in the target dataset using outputs from $G$. The network has four 3x3 convolution layers followed by 2x2 max-pooling with stride 2, one hidden fully-connected layer, and one cluster assignment layer which is also fully connected. The number of output nodes in the last layer is equal to the number of potential clusters in the target data. The output distribution is enumerated in pairs before sending to LCO. The network is randomly initialized, trained end-to-end, and optimized by stochastic gradient descent with randomly sampled 100 images per mini-batch. Note the $G$ function used by LCO is fixed during the optimization. The input data preparation is the same as above, except now the data is from $Omniglot_{eval}$. Specifically, during the training, the same mini-batch is given to both $G$ and CCN. The dense pairwise similarity predictions from $G$ are sent to LCO and are then fully utilized. The only hyper-parameter in LCO is $\sigma$, and we set it to 2 for all our experiments.

$Omniglot_{eval}$ contains 20 alphabets and each one can be used as a standalone dataset. The 20 target datasets contain a varied number (20 to 47) of characters. Therefore we can evaluate the reconstruction performance under a varied number of ground-truth clusters. We tested the situations when the number of character ($K$) in an alphabet is known and unknown. When $K$ is known, we set the target number of clusters in the clustering algorithm equal to the true number of characters. If $K$ is unknown, the common practice is to set it to a large number so that the data from different categories will not be forced to be in the same cluster. In the experiment, we merely set $K$ to 100, which is much larger than the largest dataset (47).

All constrained clustering algorithms can be used to reconstruct the semantic clusters for our problem. Since there are mis-predictions in $G$, robustness to noise is the most important factor. Here we include all four types of constrained clustering algorithms introduced in section 2 as the baselines. We provide the full set of pairwise constraints to all algorithms including ours. In other words, given an alphabet of 20 characters, it contains 400 images and 160000 predicted similarities from $G$ (note that $G$ makes predictions for both orders of a pair and for self-pairs). The full pairwise similarities were presented to all algorithms in random order, while we empirically found it has no noticeable effect on results. We pick the baseline approaches based on code availability and scalability concerning the number of pairwise constraints. Therefore we have shown results for K-means (MacQueen et al., 1967), LPNMF (Cai et al., 2009), LSC (Chen & Cai, 2011), ITML (Davis et al., 2007), SKKm (Anand et al., 2014), SKLR (Amid et al., 2016), SKMS (Anand et al., 2014), CSP (Wang et al., 2014) , and MPCK-means (Bilenko et al., 2004) as our baselines. We use the default parameters for each algorithm provided by

Table 1: Unsupervised cross-task transfer from $Omniglot_{bg}$ to $Omniglot_{eval}$. The performance is averaged across 20 alphabets which have 20 to 47 letters. The ACC and NMI without brackets have the number of clusters equal to ground-truth. The "(100)" means the algorithms use $K = 100$. The characteristics of how each algorithm utilizes the pairwise constraints are marked in the "Constraints in" column, where metric stands for the metric learning of feature representation.

| Method | Constraints in | | ACC | ACC (100) | NMI | NMI (100) |
| | Metric | Clustering | | | | |
| --- | --- | --- | --- | --- | --- | --- |
| K-means | | | 21.7% | 18.9% | 0.353 | 0.464 |
| LPNMF | | | 22.2% | 16.3% | 0.372 | 0.498 |
| LSC | | | 23.6% | 18.0% | 0.376 | 0.500 |
| ITML | o | | 56.7% | 47.2% | 0.674 | 0.727 |
| SKMS | o | | - | 45.5% | - | 0.693 |
| SKKm | o | | 62.4% | 46.9% | 0.770 | 0.781 |
| SKLR | o | | 66.9% | 46.8% | 0.791 | 0.760 |
| CSP | | o | 62.5% | 65.4% | 0.812 | 0.812 |
| MPCK-means | o | o | 81.9% | 53.9% | 0.871 | 0.816 |
| CCN (Ours) | o | o | **82.4%** | **78.1%** | **0.889** | **0.874** |

the original author except for $K$, the number of clusters. We use the same normalized images used in CCN for all algorithms.

The evaluation uses two clustering metrics. The first is normalized-mutual information (NMI) (Strehl & Ghosh, 2002) which is widely used for clustering. The second one is the clustering accuracy (ACC) (Yang et al., 2010). The ACC metric first finds the one-to-one matching between predicted clusters and ground-truth labels, and then calculates the classification accuracy based on the mapping. All data outside the matched clusters will be regarded as mis-predictions. To get high ACC, the algorithm has to generate coherent clusters where each cluster includes most of the data in a category; otherwise the score drops quickly. Therefore ACC provides better discrimination to evaluate whether the semantic clusters have been reconstructed well.

### 5.1.2 RESULTS AND DISCUSSIONS

We report the average performance over the 20 alphabets in table 1. Our approach achieved the top performance on both metrics. The CCN demonstrates strong robustness on the challenging scenario of unknown $K$. It achieved 78.1% average accuracy. Compared with 82.4% when $K$ is known, CCN has a relatively small drop. Compared to the second best algorithm, CSP, which is 65.4%, CCN outperforms it with a large gap. The classical approach MPCK-means works surprisingly well when the number of clusters is known, but its performance dropped dramatically from 81.9% to 53.9% when $K = 100$. In the performance breakdown for the 20 individual alphabets, CCN achieved 94% clustering accuracy on *Old Church Slavonic Cyrillic*, which has 45 characters (appendix table 5). Therefore the results show the feasibility of reconstructing semantic clusters using only noisy similarity predictions.

**When to use the semantic similarity?** The experiments in table 1 show a clear trend that utilizing the pairwise constraints jointly for both metric learning and minimizing the clustering loss achieves the best performance, including both MPCK-means and CCN. In the case of unknown number of clusters, where we set $K = 100$, the algorithms that use constraints to optimize clustering loss have better robustness, for example, CSP and CCN. The group that only use constraints for metric learning (ITML, SKMS, SKKm, and SKLR) significantly outperform the group that does not use it (K-means, LPNMF, LSC). However, their performance are still far behind CCN. Our results confirm the importance of jointly optimizing the metric and clustering.

**The robustness** against noisy similarity prediction is the key factor to enable the cross-task transfer framework. To the best of our knowledge, table 1 is the first comprehensive robustness comparisons using predicted constraints learned from real data instead of converting from ground-truth labels. The accuracy of $G$ in our experiment is shown in appendix table 7 and demonstrates the reasonable performance of $G$ which is on par with Matching-Net (Vinyals et al., 2016). After binarizing the

prediction at 0.5 probability, the similar pair precision, similar pair recall, dissimilar pair precision, and dissimilar pair recall among the 659 characters are $(0.392, 0.927, 0.999, 0.995)$, accordingly. The binarized predictions are better than uniform random guess $(0.002, 0.500, 0.998, 0.500)$, but are still noisy. Therefore it is very challenging for constrained clustering. The visualization of the robustness range of CCN are provided in appendix D, and shows that the robustness is related to the density of pairs involved in a mini-batch. We hypothesize that during the optimization, the gradients from wrongly predicted pairs are canceled out by each other or by the correctly predicted pairs. Therefore the overall gradient still moves the solution towards a better clustering result.

**How to predict** $K$**?** Inferring the number of clusters ($NC$) is a hard problem, but with the pairwise similarity information, it becomes feasible. For evaluation, we compute the difference between the number of dominant clusters ($NDC$) and the true number of categories ($NC^{gt}$) in a dataset. We use a naive definition for $NDC$, which is the number of clusters that have a size larger than expected size when data is sampled uniformly. In other words, $NDC = \sum_{i=1}^{K} [C_i >= E[C_i]]$, where $[\cdot]$ is an Iverson Bracket and $C_i$ is the size of cluster $i$. For example, $E[C_i]$ will be 10 if the alphabet has 1000 images and $K = 100$. Then the average difference ($ADif$) is calculated by $ADif = \frac{1}{|D|} \sum_{d \in D} |NDC_d - NC_d^{gt}|$, where $d$ (i.e., alphabet) is a dataset in $D$. A smaller $ADif$ indicates a better estimate of $K$. CCN achieves a score of 6.35 (appendix table 6). We compare this with the baseline approach SKMS (Anand et al., 2014), which does not require a given $K$ and supports a pipeline to estimate $K$ automatically (therefore we only put it into the column $K = 100$ in table 1.). SKMS gets 16.3. Furthermore, 10 out of 20 datasets from CCN's prediction have a difference between $NDC_d$ and $NC_d^{gt}$ smaller or equal to 3, which shows the feasibility of estimating $K$ with predicted similarity.

### 5.1.3 EXPERIMENTS USING THE IMAGENET DATASET

To demonstrate the scalability of our approach, we applied the same scheme on the ImageNet dataset. The 1000-class dataset is separated into 882-class ($ImageNet_{882}$) and 118-class ($ImageNet_{118}$) subsets as the random split in Vinyals et al. (2016). We use $ImageNet_{882}$ for $A$ and 30 classes ($\sim$39k images) are randomly sampled from $ImageNet_{118}$ for $T$. The difference from section 5.1.1 is that here we use Resnet-18 for both $G$ and CCN, and the weights of the backbone are pre-trained with $ImageNet_{882}$. Since the number of pairs is high and it is not feasible to feed them into other constrained clustering algorithms, we compare CCN with K-means, LSC(Chen & Cai, 2011), and LPNMF (Cai et al., 2009). We use the output from the average pooling layer of Resnet-18 as the input to these clustering algorithms. CCN gives the top performance with average ACC 73.8% when $K$ is known, and 65.2% when the number of clusters is unknown, which outperforms the second (34.5% by K-means) with a large margin. The full comparison is in appendix table 8. And the performance of $G$ is provided in appendix table 9.

## 5.2 UNSUPERVISED CROSS-DOMAIN TRANSFER LEARNING

### 5.2.1 SETUP

**Office-31** (Saenko et al., 2010) has images from 31 categories of office objects. The 4652 images are obtained from three domains: Amazon ($a$), DSLR ($d$), and Webcam ($w$). The dataset is the standard benchmark for evaluating domain adaptation performance in computer vision. We experiment with all six combinations (source $S' \rightarrow$ target $T$): $a \rightarrow w$, $a \rightarrow d$, $d \rightarrow a$, $d \rightarrow w$, $w \rightarrow a$, $w \rightarrow d$, and report the average accuracy based on five random experiments for each setting.

**The** $G$ **function** learns the semantic similarity function from the auxiliary dataset $A$, which is ImageNet with all 1000 categories. The backbone network of $G$ is Resnet-18 and has the weights initialized by ImageNet classification. The training process is the same as section 5.1.1 except the images are resized to 224.

We follow the standard protocols using deep neural networks (Long et al., 2017; Ganin et al., 2016) for unsupervised domain adaptation. The backbone network of CCN[+] is pre-trained with ImageNet. Appendix figure 7 illustrates the scheme. During the training, all source data and target data are used. Each mini-batch is constructed by 32 labeled samples from source and 96 unlabeled samples from target. Since the target dataset has no labels and could only be randomly sampled, it is crucial to have sufficient mini-batch size to ensure that similar pairs are sampled. The loss function used in our

Table 2: Unsupervised cross-domain transfer (domain adaptation) on the Office-31 dataset. The backbone network used here is Resnet-18 (He et al., 2016) pre-trained with ImageNet.

|  | A → W | D → W | W → D | A → D | D → A | W → A | Avg |
|---|---|---|---|---|---|---|---|
| Source-Only | 66.8 | 92.8 | 96.8 | 67.1 | 51.4 | 53.0 | 71.3 |
| DANN (Ganin et al., 2016) | 73.2 | 97.0 | 99.0 | 69.3 | 58.0 | 57.8 | 75.7 |
| JAN (Long et al., 2017) | 74.5 | 94.1 | **99.6** | **75.9** | 58.7 | 59.0 | 76.9 |
| CCN$^+$ (ours) | 76.7 | 97.3 | 98.2 | 71.2 | 61.0 | 60.5 | 77.5 |
| CCN$^{++}$ (with DANN) | **78.2** | **97.4** | 98.6 | 73.5 | **62.8** | **60.6** | **78.5** |

approach is equation (7) and is optimized by stochastic gradient descent. The CCN$^{+/++}$ and DANN (RevGrad) with ResNet backbone are implemented with Torch. We use the code from original author for JAN. Both DANN and JAN use a 256-dimension bottleneck feature layer.

### 5.2.2 RESULTS AND DISCUSSIONS

The results are summarized in table 2. Our approach (CCN$^+$) demonstrates a strong performance boost for the unsupervised cross-domain transfer problem. It reaches 77.5% average accuracy which gained 6.2 points from the 71.3% source-only baseline. Although our approach merely transfers more information from the auxiliary dataset, it outperforms the strong approach DANN (75.7%), and state-of-the-art JAN (76.9%). When combining ours with DANN (CCN$^{++}$), the performance is further boosted. This indicates that LCO helps mitigate the transfer problem in a certain way that is orthogonal to minimizing the domain discrepancy. We observe the same trend when using a deeper backbone network, i.e., ResNet-34. In such a case the average accuracy achieved is 77.9%, 81.1% and 82% for source-only, CCN$^+$ and CCN$^{++}$, respectively, though we used exactly the same $G$ as before (with ResNet-18 backbone for $G$). This indicates that the information carried in the similarity predictions is not equivalent to transferring features with deeper networks. More discussions are in appendix C and the performance of $G$ is provided in appendix table 11 to show that although the prediction has low precision for similar pairs ($\sim 0.2$), our approach still benefits from the dense similarity predictions.

### 5.2.3 EXPERIMENTS USING SVHN-TO-MNIST

We also evaluated the CCN$^+$ on another widely compared scenario, which uses color Street View House Numbers images (SVHN) (Netzer et al., 2011) as $S'$, the gray-scale hand-written digits (MNIST) (LeCun, 1998) as $T$. To learn $G$, we use the $Omniglot_{bg}$ as $A$. We train all the networks in this section from scratch. Our experimental setting is similar to Sener et al. (2016). We achieve the top performance with 89.1% accuracy. The performance gain from source-only in our approach is +37.1%, which wins by a large margin compared to the +23.9% of LTR (Sener et al., 2016). The full comparison is presented in appendix table 12.

## 6 CONCLUSION AND OUTLOOK

In this paper, we demonstrated the usefulness of transferring information in the form of pairwise similarity predictions. Such information can be transferred as a function and utilized by a loss formulation inspired from constrained clustering, but implemented more robustly within a neural network that can jointly optimize both features and clustering outputs based on these noisy predictions. The experiments for both cross-task and cross-domain transfer learning show strong benefits of using the semantic similarity predictions resulting in new state of the art results across several datasets. This is true even without explicit domain adaptation for the cross-domain task, and if we add a domain discrepancy loss the benefits increase further.

There are two key factors that determine the performance of the proposed framework. The first is the robustness of the constrained clustering and second is the performance of the similarity prediction function. We show robustness of CCN empirically, but we do not explore situations where learning the similarity function is harder. For example, such cases arise when there are a small number of

categories in source or a large domain discrepancy between source and target. One idea to deal with such a situation is learning G with domain adaptation strategies. We leave these aspects for future work.

## ACKNOWLEDGMENTS

This work was supported by the National Science Foundation and National Robotics Initiative (grant # IIS-1426998).

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

APPENDICES

## A    COMPARISON OF EXPERIMENTAL SETTINGS

Table 3: The list of datasets involved in each experiment. $G$ learns the similarity function from dataset A. The CCN* is optimized with dataset T or T∪S', while CCN* means CCN for cross-task transfer and CCN[+/++] for cross-domain transfer. The rows for network initialization indicate whether the network has weights initialized by training a classification task with the specified dataset. The weights are randomly initialized if not specified.

| Scheme | | Cross-Task Transfer | | Cross-Domain Transfer | |
|---|---|---|---|---|---|
| Experiment Section | | Omniglot 5.1.1 | ImageNet 5.1.3 | Office-31 5.2.1 | SVHN-MNIST 5.2.3 |
| Datasets | A | $\text{Omniglot}_{bg}$ | $\text{ImageNet}_{882}$ | $\text{ImageNet}_{1000}$ | $\text{Omniglot}_{bg}$ |
| | S' | - | - | $\text{Office-31}_{\{a,d,w\}}$ | SVHN |
| | T | $\text{Omniglot}_{eval}$ | $\text{ImageNet}_{118}$ | $\text{Office-31}_{\{a,d,w\}}$ | MNIST |
| Network Initialization | G | - | $\text{ImageNet}_{882}$ | $\text{ImageNet}_{1000}$ | - |
| | CCN* | - | $\text{ImageNet}_{882}$ | $\text{ImageNet}_{1000}$ | - |

Table 4: The list of loss functions used for training networks. The similarity prediction function (network) $G$ uses the cross-entropy (CE) loss with two classes (similar/dissimilar). The training of constrained clustering network (CCN*) involves the combinations of the learnable clustering objective (LCO), cross-entropy, and domain adaptation loss (DA).

| | LCO | CE | DA |
|---|---|---|---|
| G | | ✓ | |
| CCN | ✓ | | |
| CCN[+] | ✓ | ✓ | |
| CCN[++] | ✓ | ✓ | ✓ |

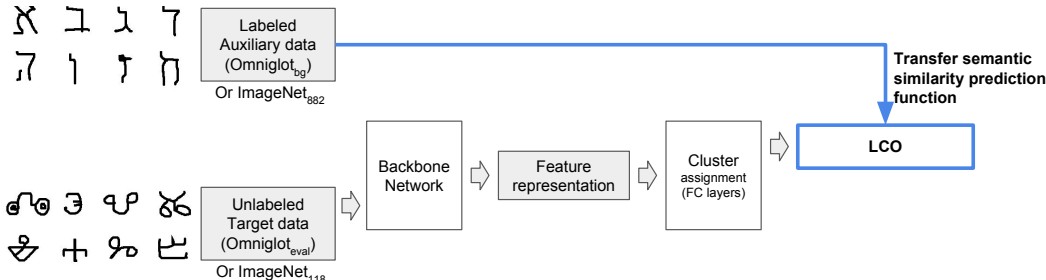

Figure 6: A diagram depicting the cross-task transfer experiment. Two experiments follow this flow: $Omniglot_{bg}$ to $Omniglot_{eval}$, and $ImageNet_{882}$ to $ImageNet_{118}$. Both have exclusive classes between source and target domain. In the ImageNet experiment, the backbone network is initialized with the weights pre-trained with $ImageNet_{882}$.

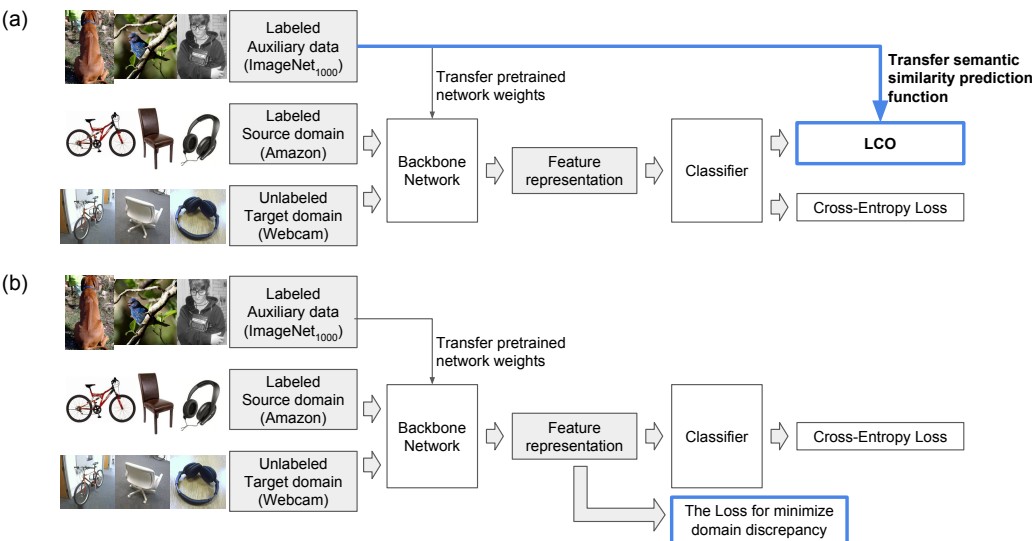

Figure 7: Comparison between domain adaptation approaches (a) Transferring semantic similarity from auxiliary data (our method), and (b) Minimizing the domain discrepancy. The diagram uses office-31 benchmark as the scenario of transferring.

## B    SUPPLEMENTARY FOR UNSUPERVISED CROSS TASK TRANSFER LEARNING

Table 5: A breakdown of the results for each alphabet in $Omniglot_{eval}$. The unsupervised cross task transfer experiment is described in section 5.1. This table shows the clustering accuracy with $K = 100$ to simulate the situation of unknown number of clusters.

| Alphabet | K-means | LPNMF | LSC | ITML | SKMS | SKKm | SKLR | CSP | MPCK-means | CCN |
|---|---|---|---|---|---|---|---|---|---|---|
| Angelic | 24% | 26% | 27% | 48% | 78% | 40% | 38% | 72% | 50% | **82%** |
| Atemayar_Qelisayer | 19% | 14% | 17% | 51% | 61% | 51% | 44% | 61% | 50% | **82%** |
| Atlantean | 19% | 15% | 19% | 57% | 72% | 55% | 51% | **77%** | 47% | 72% |
| Aurek_Besh | 23% | 18% | 21% | 45% | 49% | 44% | 45% | 76% | 71% | **90%** |
| Avesta | 22% | 18% | 19% | 47% | 29% | 39% | 38% | 65% | 52% | **76%** |
| Ge_ez | 19% | 17% | 17% | 56% | 58% | 47% | 42% | 72% | 55% | **82%** |
| Glagolitic | 22% | 19% | 21% | 42% | 38% | 58% | 62% | 66% | 61% | **85%** |
| Gurmukhi | 15% | 13% | 15% | 44% | 26% | 54% | 48% | 60% | 56% | **80%** |
| Kannada | 18% | 14% | 16% | 48% | 37% | 46% | 53% | 60% | 48% | **62%** |
| Keble | 20% | 14% | 18% | 44% | 60% | 46% | 44% | 75% | 68% | **90%** |
| Malayalam | 18% | 15% | 16% | 36% | 24% | 49% | 52% | 50% | 54% | **72%** |
| Manipuri | 17% | 15% | 16% | 49% | 40% | 53% | 54% | 66% | 62% | **85%** |
| Mongolian | 18% | 18% | 19% | 50% | 40% | 37% | 48% | 75% | 57% | **86%** |
| Old_Church_Slavonic_Cyrillic | 19% | 16% | 19% | 42% | 41% | 52% | 67% | 71% | 67% | **94%** |
| Oriya | 15% | 13% | 14% | 46% | 40% | 54% | 45% | 57% | 52% | **67%** |
| Sylheti | 14% | 13% | 14% | 49% | 32% | 35% | 37% | 59% | 43% | **64%** |
| Syriac_Serto | 20% | 18% | 19% | 55% | 70% | 42% | 35% | 70% | 47% | **73%** |
| Tengwar | 18% | 18% | 18% | 51% | 44% | 49% | 40% | 61% | 44% | **66%** |
| Tibetan | 18% | 14% | 16% | 44% | 31% | 47% | 54% | 59% | 55% | **84%** |
| ULOG | 20% | 18% | 18% | 41% | 41% | 40% | 41% | 56% | 38% | **70%** |
| Average | 19% | 16% | 18% | 47% | 46% | 47% | 47% | 65% | 54% | **78%** |

Table 6: Estimates for the number of characters across the 20 datasets in $Omniglot_{eval}$. The bold number means the prediction has error smaller or equal to 3. The $ADif$ is defined in section 5.1.2.

| Alphabet | True #class | SKMS | CCN (K=100) |
|---|---|---|---|
| Angelic | 20 | 16 | 26 |
| Atemayar_Qelisayer | 26 | 17 | 34 |
| Atlantean | 26 | 21 | 41 |
| Aurek_Besh | 26 | 14 | **28** |
| Avesta | 26 | 8 | 32 |
| Ge_ez | 26 | 18 | 32 |
| Glagolitic | 45 | 18 | **45** |
| Gurmukhi | 45 | 12 | **43** |
| Kannada | 41 | 19 | **44** |
| Keble | 26 | 16 | **28** |
| Malayalam | 47 | 12 | **47** |
| Manipuri | 40 | 17 | **41** |
| Mongolian | 30 | **28** | 36 |
| Old_Church_Slavonic_Cyrillic | 45 | 23 | **45** |
| Oriya | 46 | 22 | **49** |
| Sylheti | 28 | 11 | 50 |
| Syriac_Serto | 23 | 19 | 38 |
| Tengwar | 25 | 12 | 41 |
| Tibetan | 42 | 15 | **42** |
| ULOG | 26 | 15 | 40 |
| $ADif$ | | 16.3 | 6.35 |

Table 7: The performance of the similarity prediction function used in section 5.1. We leverage the N-way test which is commonly used in one-shot learning evaluation. The similarity is learned with $Omniglot_{bg}$ and has N-way test with $Omniglot_{eval}$ and MNIST. The experimental settings follow Vinyals et al. (2016). The raw probability output (without binarization) from our $G$ is used to find the nearest exemplar in the N-way test.

| | Omniglot-eval | | MNIST |
| Method | 5-way | 20-way | 10-way |
|---|---|---|---|
| Siamese-Nets (Koch et al., 2015) | 0.967 | 0.880 | 0.703 |
| Match-Net (Vinyals et al., 2016) | **0.981** | **0.938** | **0.720** |
| Ours | 0.979 | 0.935 | **0.720** |

Table 8: Unsupervised cross-task transfer learning on ImageNet. The values are the average of three random subsets in $ImageNet_{118}$. Each subset has 30 classes. The "ACC" has $K = 30$ while the "ACC (100)" sets $K = 100$. All methods use the features (outputs of average pooling) from Resnet-18 pre-trained with $ImageNet_{882}$ classification.

| Method | ACC | ACC(100) | NMI | NMI(100) |
|---|---|---|---|---|
| K-means | 71.9% | 34.5% | 0.713 | 0.671 |
| LSC | 73.3% | 33.5% | 0.733 | 0.655 |
| LPNMF | 43.0% | 21.8% | 0.526 | 0.500 |
| CCN (ours) | **73.8%** | **65.2%** | **0.750** | **0.715** |

Table 9: Performance of the similarity prediction function ($G$, trained with $ImageNet_{882}$) applied to three subsets of $ImageNet_{118}$. Each subset contains 30 random classes of $ImageNet_{118}$. The predictions are binarized at 0.5 to calculate the precision and recall. Random*: The expected performance when classes are uniformly distributed and make uniform random guess for similarity. It is an approximation since the number of images in each class is only roughly equal in ImageNet. We sampled 12M pairs for each set to collect the statistics. Sampling more pairs has no noticeable change to the values.

| Set | Similar Precision | Similar Recall | Dissimilar Precision | Dissimilar Recall |
|---|---|---|---|---|
| Random* | 0.033 | 0.500 | 0.967 | 0.500 |
| Set A | 0.840 | 0.664 | 0.983 | 0.994 |
| Set B | 0.825 | 0.631 | 0.981 | 0.993 |
| Set C | 0.771 | 0.671 | 0.983 | 0.990 |
| Average | 0.812 | 0.655 | 0.982 | 0.992 |

## C  Supplementary for Unsupervised Cross Domain Transfer Learning

### C.1  More discussion for the Office-31 experiments

Our experiment (table 10) shows that simply using deeper pre-trained networks produces a significant performance boost. Specifically, using Resnet-50 increases the performance 8.4 points from Alexnet, which surpasses the 6.2 points gained by using one of the state-of-the-art domain adaptation algorithms (JAN (Long et al., 2017)). If we regard pre-training with deeper networks as transferring more information, our approach is similar in this aspect since both transfer more information from the auxiliary dataset. In this case, memory limitations precluded the application of LCO to such models, and multi-GPU implementations for this problem is an area of future work.

Table 10: The performance of unsupervised transfer across domains on Office-31 dataset. The backbone networks in the comparison have different numbers of convolutional layers. AlexNet has 5 layers and the ResNets have 18~50 layers. SO is the abbreviation for source-only, which simply trains on $S'$ and directly applies the classifier on $T$. The first two rows are directly copied from Long et al. (2017). The features learned with deeper networks generalize better across domains.

| | $A \rightarrow W$ | $D \rightarrow W$ | $W \rightarrow D$ | $A \rightarrow D$ | $D \rightarrow A$ | $W \rightarrow A$ | Avg | Gain |
|---|---|---|---|---|---|---|---|---|
| AlexNet SO | 61.6 | 95.4 | 99.0 | 63.8 | 51.1 | 49.8 | 70.1 | - |
| AlexNet (JAN-A) | **75.2** | 96.6 | **99.6** | 72.8 | 57.5 | 56.3 | 76.3 | +6.2 |
| Resnet-18 SO | 66.8 | 92.8 | 96.8 | 67.1 | 51.4 | 53.0 | 71.3 | +1.2 |
| Resnet-34 SO | 73.1 | 96.4 | 98.8 | 73.8 | **63.2** | **62.1** | 77.9 | +7.8 |
| Resnet-50 SO | 74.3 | **96.8** | 98.8 | **79.5** | 61.2 | 60.6 | **78.5** | **+8.4** |

Table 11: Performance of the similarity prediction function ($G$, trained with $ImageNet_{882}$) applied on three domains of the Office-31 dataset. In total, 1.4M pairs are examined to calculate the table.

| Domain | Similar Precision | Similar Recall | Dissimilar Precision | Dissimilar Recall |
|---|---|---|---|---|
| Amazon | 0.194 | 0.696 | 0.988 | 0.894 |
| DSLR | 0.196 | 0.882 | 0.995 | 0.858 |
| Webcam | 0.213 | 0.865 | 0.994 | 0.876 |

Table 12: Unsupervised transferring across domains (S': SVHN, T: MNIST A: $Omniglot_{bg}$) without pre-trained backbone network weights. Our setup is similar to Sener et al. (2016) and Ganin et al. (2016) which therefore has a similar source-only performance.

| Method | | SVHN->MNIST | Gain |
|---|---|---|---|
| Ganin et al. (2016) | Source-Only | 54.9 | - |
| | DANN | 73.9 | +19.0 |
| Sener et al. (2016) | Source-Only | 54.9 | - |
| | LTR | 78.8 | +23.9 |
| Saito et al. (2017) | Source-Only | 70.1 | - |
| | ATDA | 86.2 | +16.1 |
| Tzeng et al. (2017) | Source-Only | 60.1 | - |
| | ADDA | 76.0 | +15.9 |
| Ours | Source-Only | 52.0 | - |
| | CCN$^+$ | **89.1** | **+37.1** |

Table 13: Performance of the similarity prediction function ($G$, trained with $Omniglot_{bg}$) applied on the MNIST dataset. In total, 5M pairs are sampled to calculate the table.

|  | Similar Precision | Similar Recall | Dissimilar Precision | Dissimilar Recall |
|---|---|---|---|---|
| Random | 0.100 | 0.500 | 0.900 | 0.500 |
| MNIST | 0.782 | 0.509 | 0.946 | 0.984 |

# D ROBUSTNESS ANALYSIS OF CONSTRAINED CLUSTERING NETWORK

## D.1 SETUP

To quickly explore the large combination of factors that may affect the clustering, we use a small dataset (MNIST) and a small network which has two convolution layers followed by two fully connected layers. The MNIST dataset is a dataset of handwritten digits that contains 60k training and 10k testing images with size 28x28. Only the training set is used in this section and the raw pixels, which were normalized to zero mean and unit standard deviation, were fed into networks directly.

The networks were randomly initialized and the clustering training was run five times under each combination of factors; we show the best final results, as is usual in the random restart regime. The mini-batch size was set to 256, thus up to 65536 pairs were presented to the LCO per mini-batch if using full density (D=1). There were 235 mini-batches in an epoch and the optimization proceeded for 15 epochs. The clustering loss was minimized by stochastic gradient descent with learning rate 0.1 and momentum 0.9. The predicted cluster was assigned at the end by forwarding samples through the clustering networks. The best result in the five runs was reported.

To simulate different performance of the similarity prediction, the label of pairs were flipped according to the designated recall. For example, to simulate a 90% recall of similar pair, 10% of the ground truth similar pair in a mini-batch were flipped. The precision of similar/dissimilar pairs is a function of the recall of both type of pairs, thus controlling the recall is sufficient for the evaluation. The recalls for both similar and dissimilar pairs were gradually reduced from one to zero at intervals of 0.1.

## D.2 DISCUSSION

The resulting performance w.r.t different values of recall, density, and number of clusters is visualized in Figure 8. A bright color means high NMI score and is desired. The larger the bright region, the more robust the clustering is against the noise of similarity prediction. The ACC score shows almost the same trend and is thus not shown here.

**How does similarity prediction affect clustering?** Looking at the top-left heat map in figure 8, which has $D = 1$ and 10 clusters, it can be observed that the NMI score is very robust to low similar pair recall, even lower than 0.5. For recall of dissimilar pairs, the effect of recall is divided at the 0.5 value: the clustering performance can be very robust to noise in dissimilar pairs if the recall is greater than 0.5; however, it can completely fail if recall is below 0.5. For similar pairs, the clustering works on a wide range of recalls when the recall of dissimilar pairs is high.

In practical terms, robustness to the recall of similar pairs is desirable because it is much easier to predict dissimilar pairs than similar pairs in real scenarios. In a dataset with 10 categories e.g. Cifar-10, we can easily get 90% recall for dissimilar pairs with purely random guess if the number of classes is known, while the recall for similar pairs will be 10%.

**How does the density of the constraints affect clustering?** We argue that the density of pairwise relationships is the key factor to improving the robustness of clustering. The density $D = 1$ means that every pair in a mini-batch is utilized by the clustering loss. For density $D = 0.1$, it means only 1 out of 10 possible constraints is used. We could regard the higher density as better utilization of the pairwise information in a mini-batch, thus more learning instances contribute to the gradients at once. Consider a scenario where there is one sample associated with 5 true similar pairs and 3 false similar pairs. In such a case, the gradients introduced by the false similar pairs have a higher chance to be overridden by true similar pairs within the mini-batch, thus the loss can converge faster and is less

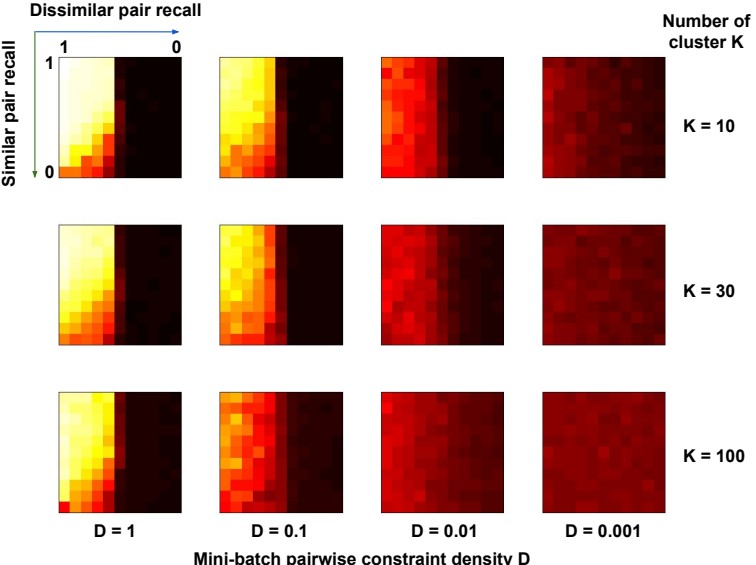

Figure 8: The clustering performance with different pairwise density and number of clusters. A bright color means that the NMI score is close to 1 while black corresponds to 0. The density is defined as a ratio compared to the total number of pair-wise combinations in a mini-batch. The number of clusters defines the final softmax output dimensionality. In each sub-figure, we show how the scores change w.r.t. the similar pair recall and dissimiliar pair recall.

affected by errors. In Figure 8, we could see when density decreases, the size of the bright region shrinks significantly.

In our implementation, enumerating the full pairwise relationships introduces negligible overhead in computation time using GPU. Although there is overhead for memory consumption, it is limited because only the vector of predicted distributions has to be enumerated for calculating the clustering loss.

**The effect of varying the number of Clusters** In the MNIST experiments, the number of categories is 10. We augment the softmax output number up to 100. The rows of figure 8 show that even when the number of output categories is significant larger than the number of true object categories, e.g. $100 > 10$, the clustering performance NMI score only degrades slightly.

