# OpenReview forum: "Learning to cluster in order to transfer across domains and tasks"
_ICLR.cc/2018/Conference — Accept (Poster)_

### Official Review · AnonReviewer1 · 2017-11-27

**Rating:** 5
**Confidence:** 4

**Review:**

(Summary)
This paper tackles the cross-task and cross-domain transfer and adaptation problems. The authors propose learning to output a probability distribution over k-clusters and designs a loss function which encourages the distributions from the similar pairs of data to be close (in KL divergence) and the distributions from dissimilar pairs of data to be farther apart (in KL divergence). What's similar vs dissimilar is trained with a binary classifier.

(Pros)
1. The citations and related works cover fairly comprehensive and up-to-date literatures on domain adaptation and transfer learning.
2. Learning to output the k class membership probability and the loss in eqn 5 seems novel.

(Cons)
1. The authors overclaim to be state of the art. For example, table 2 doesn't compare against two recent methods which report results exactly on the same dataset. I checked the numbers in table 2 and the numbers aren't on par with the recent methods. 1) Unsupervised Pixel-Level Domain Adaptation with Generative Adversarial Networks, Bousmalis et al. CVPR17, and 2) Learning Transferrable Representations for Unsupervised Domain Adaptation, Sener et al. NIPS16. Authors selectively cite and compare Sener et al. only in SVHN-MNIST experiment in sec 5.2.3 but not in the Office-31 experiments in sec 5.2.2.
2. There are some typos in the related works section and the inferece procedure isn't clearly explained. Perhaps the authors can clear this up in the text after sec 4.3.

(Assessment)
Borderline. Refer to the Cons section above.

---

> ### Author Response · Authors · 2017-12-31
> **Rebuttal**
>
> Thank you for the detailed comments. We would like to respond to the cons you mentioned.
>
> 1. First, we discuss several points about the cross domain transfer experiment. We separate them in the following:
>
> (1) Thank you so much for referring us to Bousmalis’s CVPR work. It is a great work and we added the citation in our updated version. However, that paper doesn’t include the experiments on Office-31 nor SVHN-MNIST. Could you please share the mentioned results with us?
>
> (2) Sener’s paper is another great work. However, to make the numbers be comparable, the backbone networks (e.g., AlexNet, VGG, ResNet…), training configurations, and image preprocessing must be the same. Our table 2 for Office-31 makes sure the setup is comparable. Sener’s work has a different setup and has not released the code yet. Therefore a fair comparison on Office-31 dataset is not viable at this date. For the SVHN-to-MNIST experiment, we use another way to compare the results. In table 10, it shows the relative performance gain against the baselines (source-only) of each original paper. In such way we don’t need the code of the referred approaches be available to make the absolute numbers comparable.
>
> (3) Based on above explanation, our statement about the performance achievement is still valid.
>
> (4) We would like to heavily emphasize that in the context of domain adaptation, our work should be considered as a complementary strategy to current mainstream domain adaptation approaches since the proposed LCO does not minimize the domain discrepancy. Instead of beating the benchmark, the main purpose of our paper is showing that transferring more information is an effective strategy for transfer learning across both tasks and domains. In fact, we show improved results using one of the domain adaptation methods (DANN) but we anticipate that incorporating later advances in domain adaptation using discrepancy losses will improve this. Furthermore, the performance of the framework is largely dependent on the accuracy of the learned similarity prediction. We use a naive implementation of a similarity prediction network (SPN) to obtain the presented numbers. The performance can be improved further when a more powerful similarity learning algorithm is available.
>
> 2. The inference procedure for cross task transfer applies forward propagation on the network in figure 4 and uses the outputs after the cluster assignment layer. For cross domain transfer, the network is in figure 5 and the inference outputs is from the classification layer. We included the description in the updated version. Thank you so much for the feedback to improve the clarity.

---

### Official Review · AnonReviewer2 · 2017-11-28
**New method with a rough presentation**

**Rating:** 7
**Confidence:** 4

**Review:**

The authors propose a method for performing transfer learning and domain adaptation via a clustering approach. The primary contribution is the introduction of a Learnable Clustering Objective (LCO) that is trained on an auxiliary set of labeled data to correctly identify whether pairs of data belong to the same class. Once the LCO is trained, it is applied to the unlabeled target data and effectively serves to provide "soft labels" for whether or not pairs of target data belong to the same class. A separate model can then be trained to assign target data to clusters while satisfying these soft labels, thereby ensuring that clusters are made up of similar data points.

The proposed LCO is novel and seems sound, serving as a way to transfer the general knowledge of what a cluster is without requiring advance knowledge of the specific clusters of interest. The authors also demonstrate a variety of extensions, such as how to handle the case when the number of target categories is unknown, as well as how the model can make use of labeled source data in the setting where the source and target share the same task.

The way the method is presented is quite confusing, and required many more reads than normal to understand exactly what is going on. To point out one such problem point, Section 4 introduces f, a network that classifies each data instance into one of k clusters. However, f seems to be mentioned only in a few times by name, despite seeming like a crucial part of the method. Explaining how f is used to construct the CCN could help in clarifying exactly what role f plays in the final model. Likewise, the introduction of G during the explanation of the LCO is rather abrupt, and the intuition of what purpose G serves and why it must be learned from data is unclear. Additionally, because G is introduced alongside the LCO, I was initially misled into understanding was that G was optimized to minimize the LCO. Further text explaining intuitively what G accomplishes (soft labels transferred from the auxiliary dataset to the target dataset) and perhaps a general diagram of what portions of the model are trained on what datasets (G is trained on A, CCN is trained on T and optionally S') would serve the method section greatly and provide a better overview of how the model works.

The experimental evaluation is very thorough, spanning a variety of tasks and settings. Strong results in multiple settings indicate that the proposed method is effective and generalizable. Further details are provided in a very comprehensive appendix, which provides a mix of discussion and analysis of the provided results. It would be nice to see some examples of the types of predictions and mistakes the model makes to further develop an intuition for how the model works. I'm also curious how well the model works if, you do not make use of the labeled source data in the cross-domain setting, thereby mimicking the cross-task setup.

At times, the experimental details are a little unclear. Consistent use of the A, T, and S' dataset abbreviations would help. Also, the results section seems to switch off between calling the method CCN and LCO interchangeably. Finally, a few of the experimental settings differ from their baselines in nontrivial ways. For the Office experiment, the LCO appears to be trained on ImageNet data. While this seems similar in nature to initializing from a network pre-trained on ImageNet, it's worth noting that this requires one to have the entire ImageNet dataset on hand when training such a model, as opposed to other baselines which merely initialize weights and then fine-tune exclusively on the Office data. Similarly, the evaluation on SVHN-MNIST makes use of auxiliary Omniglot data, which makes the results hard to compare to the existing literature, since they generally do not use additional training data in this setting. In addition to the existing comparison, perhaps the authors can also validate a variant in which the auxiliary data is also drawn from the source so as to serve as a more direct comparison to the existing literature.

Overall, the paper seems to have both a novel contribution and strong technical merit. However, the presentation of the method is lacking, and makes it unnecessarily difficult to understand how the model is composed of its parts and how it is trained. I think a more careful presentation of the intuition behind the method and more consistent use of notation would greatly improve the quality of this submission.

=========================
Update after author rebuttal:
=========================
I have read the author's response and have looked at the changes to the manuscript. I am satisfied with the improvements to the paper and have changed my review to 'accept'.

---

> ### Author Response · Authors · 2017-12-31
> **Great suggestion**
>
> We greatly appreciate your constructive suggestions for enhancing the clarity. The suggestions make sense and can be done with minor refinement. We adopt them in our updated version. Thank you so much for your contribution.
>
> The experimental settings you mentioned are also very interesting to us. Although we believe the current experimental settings serve well the purpose of supporting the main claim that transferring predictive similarity is an effective and generic way of transfer learning, the settings you described will help exploring other dimensions of the framework. For example, learning the similarity with only limited semantic categories, i.e., only use T’ instead of the large A. We would like to include the aspects you mentioned in a future work.
>
> For the discussion about the Office-31 experiment, we are pleased to explain why the ImageNet data (or A) doesn’t need to be presented during the training with T. The community uses the weights pretrained with ImageNet as a generic initialization due to its training on a large number of categories. We leverage a similar idea and argue that a semantic similarity learned with ImageNet-scale probably generalize well to unseen classes. Our cross-task ImageNet experiment shows support to this idea. Thereby we augment the transferring scheme from only transferring the weights to transferring both weights and the learned similarity prediction. In other words, once the similarity prediction network is learned on ImageNet, it can be applied directly to other image classification datasets without access to ImageNet dataset, just like how we use the pre-trained weights (features).

---

### Official Review · AnonReviewer3 · 2017-11-30
**The authors propose a novel framework for task-/domain- transfer in an unsupervised setting (no labels for target data). The idea of defining the similarity based-clustering within domain adaptation /transfer learning framework is novel. Experiments are thorough and show clear benefits of the proposed learning strategy.**

**Rating:** 9
**Confidence:** 5

**Review:**

pros:
This is a great paper - I enjoyed reading it. The authors lay down a general method for addressing various transfer learning problems: transferring across domains and tasks and in a unsupervised fashion. The paper is clearly written and easy to understand. Even though the method combines the previous general learning frameworks, the proposed algorithm for  LEARNABLE CLUSTERING OBJECTIVE (LCO) is novel, and fits very well in this framework.  Experimental evaluation is performed on several benchmark datasets - the proposed approach outperforms state-of-the-art for specific tasks in most cases.

cons/suggestions:
- the authors should discuss in more detail the limitations of their approach: it is clear that when there is a high discrepancy between source and target domains, that the similarity prediction network can fail. How to deal with these cases, or better, how to detect these before deploying this method?
- the pair-wise similarity prediction network can become very dense: how to deal with extreme cases?

---

> ### Author Response · Authors · 2017-12-31
> **Discussion**
>
> Thank you for your comments and for recognizing the work. We are pleased to have a discussion of the limitations here and we added part of it into the paper.
>
> As you pointed out, the performance of the similarity prediction is crucial, and the amount of performance gain on the target task is proportional to the accuracy of the similarity. One idea for enhancement is applying domain adaptation to learn the similarity prediction network (SPN). Any existing adaptation method can be used to train the SPN; as long as that adaptation method can deal with the high discrepancy between source and target, our framework will directly benefit from improvements in the learned SPN. With our proposed learning framework, the meta-knowledge carried by the SPN can then be transferred across either tasks or domains.
>
> An extreme case involving the density of pairs is when there is a large number of categories in the target dataset. If there are 100 categories and we only randomly sample 32 instances for a mini-batch, there could be no similar pairs in a mini-batch since the instances are all from different categories. The LCO might not work well in this case since it has a form of contrastive loss. There are two ways to address this. The first is enlarge the mini-batch size, so that the number of sampled similar pairs increases. This method is limited by the memory size. The second way is to obtain dense similarity predictions offline. Then a mini-batch is sampled based on the pre-calculated dense similarity to ensure similar pairs are presented.

---

### Public Comment · (anonymous) · 2017-11-23
**Alternatives for Backbone network**

Great work! Thank you for your contribution and I have three questions.
a) Do you have any suggestion for situations when a pre-trained backbone network is not available, it seems very important for getting good results. As far as I understand, training backbone end-to-end in the proposed solution would not be easy due to G's dependence over it.
b) What range of values for sigma do you recommend? In paper you used fixed value i.e. 2 for all experiments.
c) In actual implementation of equation 1 and 3, do you add terms D_KL(P* || Q) + D_KL(Q* || P) (or 2 x D_KL(P* || Q) as they are symmetric or is it fine to just optimize one e.g. D_KL(P* || Q)?

---

> ### Author Response · Authors · 2017-11-26
> **Question Answering**
>
> Thank you very much for your interest. The responses of each point are in below:
> a) It dependents on how hard the dataset is. In our experiments, the backbone networks and G are randomly initialized for the experiments on Omniglot and MNIST, and they perform well. For the experiments on Office-31 and the subsets of ImageNet, we do initialize the backbones with pre-training. When dealing with real-world photos, it is a common practice of pre-training, especially when the target dataset is small, e.g., Office-31. The general suggestion is: If the dataset needs a pre-trained network to help it performs well on classification, then it would be better also to use a pre-trained network in our approach.
> b) We believe the fixed value 2 is sufficient for most of the case. Our experiments involve the datasets of different complexity (e.g. MNIST vs ImageNet), unbalanced dataset (e.g. Office-31), and the varied number of categories (e.g. Omniglot alphabets). It shows the same setting performs well on the diverse conditions. In practice, we do see sometimes the performance could be improved by setting a larger margin (e.g. 2~5), but that extra gain seems dataset-dependent. Therefore we use the fixed value 2 as a conservative but universal setting.
> c) As you mentioned, it is for making the distance metric symmetric. Using only one part will introduce a hard question: Which one should be chosen? I have no clear answer for this. But from the implementation aspect, the symmetric form has an efficient vectorization thus it adds neglectable computational time compared to using only one part.

---

> > ### Public Comment · (anonymous) · 2017-11-29
> > **About cross-domain transfer, Figure 5**
> >
> > Did you train the model depicted in Figure 5 end-to-end including backbone with classification model (except for G)?
> > What are your thoughts on the applicability of LCO for cross-domain transfer in fields other than vision and language modelling?

---

> > > ### Author Response · Authors · 2017-12-22
> > > **Performance is dependent on G**
> > >
> > > Yes, figure 5 is trained end-to-end, except G.
> > > The applicability of LCO is decided by its learnable part (G). In our experiment (appendix D), if G can perform better than a random guess, the clustering could benefit from it. Therefore the limitation would be how to learn a G, so its prediction of similarity is better than a random guess. If the datasets have high domain discrepancy, learn the G with domain adaptation strategy will be a good idea.

---

### Author Response · Authors · 2018-01-04
**Paper updated**

The paper is updated. We added part of the discussions raised by AnonReviewer3 and enhanced the clarity suggested by AnonReviewer2 and AnonReviewer1. We greatly appreciate the contribution of reviewers.

---

### Decision · Program_Chairs · 2018-01-29
**ICLR 2018 Conference Acceptance Decision**

**Decision:**

Accept (Poster)

**Comment:**

Pros
-- A novel formulation for cross-task and cross-domain transfer learning.
-- Extensive evaluations.

Cons
-- Presentation a bit confusing, please improve.

The paper received positive reviews from reviewers. But the reviewers pointed out some issues with presentation and flow of the paper. Even though the revised version has improved, the AC feels that it can be improved further. For example, as pointed out by reviewers, different parts of the model are trained using different losses and / or are pre-trained. It would be worth clarifying that. It might help if the authors include a pseudocode / algorithm block to the final version of the paper.